# A One-Health Model for Reversing Honeybee (*Apis mellifera* L.) Decline

**DOI:** 10.3390/vetsci7030119

**Published:** 2020-08-27

**Authors:** Philip Donkersley, Emily Elsner-Adams, Siobhan Maderson

**Affiliations:** 1Lancaster Environment Centre, Lancaster University, Lancaster LA14YQ, UK; 2Elsner Research and Consulting, 8118 Zürich, Switzerland; emilyadams13@gmail.com; 3Department of Geography & Earth Sciences, Aberystwyth University, Aberystwyth SY233FL, UK; sim30@aber.ac.uk

**Keywords:** honeybee, *Apis mellifera*, One-Health, nexus, landscape, beekeeper, pathogens

## Abstract

Global insect decline impacts ecosystem resilience; pollinators such as honeybees (*Apis mellifera* L.) have suffered extensive losses over the last decade, threatening food security. Research has focused discretely on in-hive threats (e.g., *Nosema* and *Varroa destructor*) and broader external causes of decline (e.g., agrochemicals, habitat loss). This has notably failed to translate into successful reversal of bee declines. Working at the interdisciplinary nexus of entomological, social and ecological research, we posit that veterinary research needs to adopt a “One-Health” approach to address the scope of crises facing pollinators. We demonstrate that reversing declines will require integration of hive-specific solutions, a reappraisal of engagement with the many stakeholders whose actions affect bee health, and recontextualising both of these within landscape scale efforts. Other publications within this special issue explore novel technologies, emergent diseases and management approaches; our aim is to place these within the “One-Health” context as a pathway to securing honeybee health. Governmental policy reform offers a particularly timely pathway to achieving this goal. Acknowledging that healthy honeybees need an interdisciplinary approach to their management will enhance the contributions of veterinary research in delivering systemic improvements in bee health.

## 1. Introduction

Honeybee (*Apis mellifera* L.) populations have been in decline in many parts of Europe and North America since the start of the millennium [1,2]. Much of the discussion around bee decline has focused on external factors like chemical pesticides, land use change, diseases and pests [3]. Other internal physiological traits such as host gut microbiota, immune response or genetic variation have also been implicated [4,5,6]. To date the vast majority of this research was carried out by entomologists, physiologists and ecologists, but not veterinary researchers. Veterinary engagement with honeybees has been limited, save for a handful of papers focusing on disease incidence and management [7,8]. As this special issue highlights, veterinary science is increasingly required to engage with honeybee health. Adopting an interdisciplinary approach both for practical animal health purposes like prescribing antibiotics for use in hives because they are food producing animals [9], for food security and because honeybees are an indicator species for the wider health of the environment [10,11]. Here we explore the potential for veterinary science to rise to the challenge of ongoing bee declines through the One-Health approach.

The One-Health concept has been discussed since the mid-2000s, as an initiative between the medical and veterinary fields, acknowledging the need for a greater focus on zoonoses in the wake of the SARS outbreak and avian influenza outbreaks. Adopted by the WHO, OIE and FAO, as well as veterinary and medical societies around the world [12], the concept emphasises the interconnectedness of human, animal and environmental health, noting in particular that human health depends on a healthy and functioning ecosystem [13]. Although there are discussions about the different terms describing this interconnectedness (for example ‘One-Health’, ‘EcoHealth’, ‘Planetary Health’, [14]), they share a common focus on encouraging disciplines like medicine, veterinary science, ecological and environmental science to start collaborating across disciplinary boundaries. Although the focus has been on zoonotic diseases, the One-Health approach emphasises a holistic understanding to tackle challenges [13], and is therefore highly relevant for discussions about honeybee health. 

Honeybees are closely linked to human well-being, through pollination of wild and agricultural plants, and through honey production [15]. They provide an important example of the need for interdisciplinary engagement over health, as their health is determined by many factors within the landscape, within human society and within the hive. In the UK, where the research underpinning this article took place, honeybees are primarily managed by beekeepers: wild colonies predominantly died out from the 1980s following the invasion of the infectious mite *Varroa destructor* [16], although there are still feral honeybee colonies present in the landscape [17]. The UK landscape is heavily agricultural, with arable and grazing land use covering 72% of the landscape. Honeybees co-exist with diseases and pest species, ranging from viruses to bacteria and Acari (See Table A1). 

In this regard, the co-existence of disease-causing organisms and a species of human importance is not unusual—all domestic animals, and indeed humans, have a high incidence of potentially pathogenic organisms on their body [18]. What is important from a health perspective is understanding the relationships between the species present, and the context in which they are co-habiting. Hinchliffe et al. [19] argue that farm animal health is not a simple binary between healthy and unhealthy, where the definition of ‘healthy’ implies an absence of disease, but rather a result of the interactions between host, micro-organisms, environment, host immune system, management practices, and so forth [19]. The distinction between healthy and unhealthy is perhaps better described as a ‘borderland’ rather than a ‘borderline’, acknowledging that outbreaks of disease are rarely the result of incursions of a parasite or pest from outside into a ‘healthy’ area, but rather caused by a shift in the relationships between host, microorganism, management practices, host immune system and so on. This understanding of disease emphasises that diseases are endemic and co-generated [20].

Hinchliffe [20] suggests that addressing animal health issues requires a transdisciplinary approach to understanding disease, drawing in particular on knowledge from veterinarians, stock-people and farmers. Such an understanding of disease is vital to understanding the concept of health in honeybees. This paper explores three different groups of factors, operating at different scales, which, independently or in combination, can tip the balance of bee health into a condition of ‘healthy’ or ‘unhealthy’. We consider how bee health is managed in hive-scale factors, human-scale factors and landscape-scale factors (Figure A1). Following critiques of One-Health initiatives to date [13,21,22], we actively engage with the environmental factors affecting honeybee health, and with the social science analyses [23]. We conclude by reflecting on the future for honeybee health, arguing that a One-Health approach could provide a useful framework for the many, diverse researchers, policy-makers and practitioners currently seeking to support healthy honeybee populations.

## 2. Materials and Methods 

This paper draws on three projects on honeybee health that all used mixed methods, and were inspired by a transdisciplinary research perspective (Figure A1). Mixed methods research engages with multiple data types and different worldviews; our research suggests that this is necessary for improving bee health. Authors 1 and 2 worked in Lancashire, UK, with local hobbyist beekeepers, and in Herefordshire, UK, with a commercial beekeeper. Author 3 worked across the UK with commercial and hobby beekeepers, many of whom were nationally respected writers and lecturers on bee health. Authors 1 and 3 also interviewed policy-makers, scientific and academic experts, and bee health experts. Methods used in the authors’ research included interviews, participant observation [24], in-hive and laboratory experiments [25], archival analysis and document analysis [26]. Archives and documents can provide temporal and/or contextual richness to otherwise singular understandings of bee health and decline [27,28]. By drawing together research and understandings about bee health from these diverse perspectives, the authors are well placed to present bee health in its full complexity.

## 3. Hive-Scale Factors

Most direct human–honeybee contact occurs at the hive, which is a highly complex environment. Honeybee hives are rich in bacterial, viral and insect diversity, co-existing within and on honeybees (Table A1). Beekeeping practices oriented towards health management and ‘biosecurity’ are often focused on individual beekeepers’ hives, and those within flying range of the bees—up to 10 km from the hive. Security against diseases and pests has predominantly relied on creating barriers and spatial separations between livestock, pathogens and vectors of pathogens, and dividing them into categories such as healthy and diseased and then endeavouring to control movement across barriers [12]. Barriers include physical management practices, for example the use of disinfectant to stop pathogens being transported on clothes or vehicles, and legal/policy barriers, including legislation on movement, and surveillance of at-risk sites or animals [29]. Current beekeeping biosecurity advice focuses on good husbandry to maintain strong colonies that can resist infection or invasion [30], and specific management techniques designed to avoid the arrival of, or spread of, diseases, i.e., techniques that create a barrier around colonies and apiaries, known as ‘barrier management methods’.

However, honeybees pose three specific management challenges to beekeepers and others tasked with monitoring and managing their health. First, honeybees are semi-domesticated animals: though beekeepers manage honeybee colonies, they are reliant on the open environment for large parts of their lives. Foraging excursions are inherently hazardous: bees interact with individuals from other colonies, and potentially transmit diseases and pests by arriving back to the wrong hive [31]; robbing other hives for food [32]; visiting plants previously visited by infected bees [33], and simply through life history traits such as mass mating where many individuals from different hives congregate at once [34]. These communal behaviours emphasise the challenges facing beekeepers and others responsible for healthy honeybee populations, and make clear the need for more interdisciplinary, flexible approaches to understanding and managing health. 

Second, unusually in a veterinary context, the unit of management and treatment in the case of honeybees is the colony, not the individual animal—with colonies being composed of between 10,000 and 40,000 individuals, depending on the time of year. Honeybee colonies are highly tolerant of diseases: their behaviour is characterised by temporal polyethism, or age-dependent division of labour [35], and the age at which they transition between tasks depends on the condition of the colony and the individuals. Bees suffering from disease outbreaks or other physical challenges transition more quickly than healthy bees through to the most hazardous tasks such as out-of-hive foraging, reflecting the shorter longevity of these tasks [36]. An example where this is an issue is the parasitic mite *Varroa destructor*, which has a well-documented negative effect on honeybees. Its direct parasitism causes loss of haemolymph and body fat, reduced longevity and other issues, and it is a vector for viruses (Table A1), which then further damage the individual bees [37]. There are treatments available for *Varroa*, although none are perfect and some can also weaken bees [38]. 

Managing Varroa and its effects on honeybees raises the third challenge: honeybee managers are seeking to manage a species with which we share few characteristics. Even for mammalian livestock, it is difficult to measure what animals experience, although veterinarians and animal scientists have developed a series of parameters to measure nutrition, health, physical comfort and behavior, which can be interpreted to give an idea of the mental state of an animal [39]. Communication with insects is even more challenging, and although some honeybee communications have been decoded [40], judging the physical and emotional states of honeybees remains a challenge. Parasite and pathogen challenges like the negative direct and indirect effects of *Varroa*, require beekeepers to be skilled observers of their colonies [24], using proxy measures within the hive such as amount of brood as signs of stress [41]. 

Managing bee health therefore involves a constant process of negotiation between actions that might benefit a colony and actions that might put it at risk. Rather than treating all pathogen and pest species present within a single hive, it may be more useful to work within a conceptual framework of “tipping points”, where attention is focused only on levels of a disease or pest that pose sufficient threat [20]. This more holistic approach to health management is somewhat practiced within the beekeeping community under the term “integrated pest management” (IPM). IPM within a beekeeping context pushes for intense monitoring of pests and parasites, allowing action beyond prescribed tipping points of symptoms or density of parasites. 

Beekeepers, and bee inspectors charged with managing bee health adopt IPM techniques that seek to treat all aspects of bee lives, rather than working in isolation on single pathogen or pest species. IPM retains a hive-level focus, and does not engage directly with factors operating at a landscape-scale, such as forage availability or pesticide use, nor with economic issues like trade in bees (a classic source of infection of novel parasites and pathogens). Beekeepers and inspectors seeking to create resilient, healthy colonies therefore struggle to create the ideal conditions for healthy bees—one with plentiful, nutritious food across a season, and with limited chemicals, and where threats of novel pests and parasites are kept to a minimum. 

## 4. Human-Scale Factors

Many of the factors causing bee decline occur within the hive. Beekeepers are responsible for monitoring hive health, as well as liaising with others whose actions affect bees, including farmers and bee inspectors. Beekeepers’ knowledge and practices are central to any efforts to reverse bees’ decline. The role of beekeepers in monitoring and ensuring honeybee health serves as a linkage between hive-based factors, and wider landscape scale factors affecting bees. Beekeepers’ observations must be a cornerstone of any efforts to maintain species wellbeing, if they are to succeed [42,43]. There is often a tension between scientific assessment of bee health, and beekeepers’ practical engagement with their colonies. This is rooted in fundamentally different ways of gaining knowledge about bees and the wider world [44]. Much of our current knowledge of bee health is generated via an epidemiological model, while beekeepers’ knowledge is the result of a highly situated, locally generated knowledge [45]. This combines practical husbandry and direct observation, often resulting from generations of beekeeping in the same local landscape, as well as the more formal scientific knowledge that is the cornerstone of veterinary analysis of animal health. Beekeepers note how their practice results in them ‘seeing like a bee’ (e.g., interpreting the landscape’s challenges and opportunities in terms of their colonies) [26]. 

Within the UK, national bee inspectors are tasked with inspecting colonies for foulbrood diseases, as well as providing beekeepers with information and education on bee health. While the bee inspectorate is overseen by the governmental Animal and Plant Health Agency (APHA), and inspections are carried out according to strict epidemiological standards, which are primarily focused on observing colonies for foulbrood diseases (Table A1), the education and outreach aspect of the position is rooted in personal working relationships with individual beekeepers. 

While beekeepers are responsible for the immediate health of their bees, some beekeeping practices may be ultimately contrary to the short, and long term, health of bees. Characteristics of bees that may be attractive to beekeepers, such as high productivity, a low tendency to swarm, and being easy to handle and manipulate, may be ultimately counter to bees’ welfare [41,46]. Over the past decade, concern about the negative effects of various in-hive treatments for *Varroa* has led to an increase in beekeepers practicing treatment-free beekeeping [47,48]. This approach can be highly problematic for the wider beekeeping community, as *Varroa* infestation has been widely associated with colony death [49], and the spread of infection between neighbouring colonies through robbing and drifting [50], or even to non-*Apis* pollinators [51] (see Section 6).

Hobby and professional beekeepers often have opposing perspectives on what are acceptable risks to their bees, and the timescale and nature of necessary human interventions. While most beekeepers express a deep sense of stewardship and responsibility to their bees’ welfare, they have differing interpretations of what this involves [49]. For example, some beekeepers carry out their own bee breeding programmes, in an effort to move away from a reliance on chemical treatments for *Varroa* [52]. Although the treatment of *Varroa* is a volatile debate amongst the beekeeping community [49], the question of how beekeepers can manage *Varroa* whilst minimising the damaging side effects of miticides [53] highlights the human scale of factors affecting bee health. While beekeepers stay informed of innovative developments in veterinary understandings of bee health, and new in-hive treatments and management practices to combat the relentless onslaught of various infections [26], they also note the inadequacy of relying on such an approach. Others argue that their own husbandry can only go so far; ultimately, systemic change in landscape management is necessary to ensure bee health [26]. To this end, some beekeepers focus on negotiating with other land managers, upon whose land bees may be placed, and often educating them about best practice for pollinators. Some interviewees noted what they perceived as a discrepancy between land managers’ formal adherence to agri-environment schemes designed to enhance forage and habitat for bees, and actual significant landscape enhancement. Incorporating the locally situated knowledge of beekeepers into land management strategy will be necessary to assure bee health.

Formal scientific understanding of bee health is primarily relied upon by policy-makers in efforts to address pollinator decline, with beekeepers’ experiential and observational knowledge often being dismissed as anecdotal [54]. Although it can be challenging to incorporate such diversity of views, it is central to successful conservation movements [55]. As bee decline affects biodiversity, strategies from other conservation efforts are relevant [56,57]. Scientific assessments of landscape effects have successfully been applied alongside beekeepers’ knowledge, leading to a broader, more nuanced understanding of complex environmental synergies [46]. A One-Health perspective facilitates constructive engagement with seemingly disparate understandings of bee health, and enables the insights of the full, heterogeneous beekeeping community, as well as bee scientists, veterinarians, and farmers, to be combined. By considering bee decline within a One-Health framework, we can also address the shared challenges to human, bee, and wider environmental health created by the industrial food system [58]. It is clear that in-hive challenges, which are observed by both beekeepers, and by qualified external observers such as bee inspectors, are exacerbated by challenges in the wider environment [1,59]. The next section considers the landscape-level factors affecting bee health, and how the colony itself can and does work to avoid ill health.

## 5. Landscape-Scale Factors

A number of factors relevant to honeybee health operate at a landscape-scale. These factors are very large-scale, hard to see, slow-moving changes with an inherently shifting baseline [59]. We explore two key landscape-scale factors in the rest of this section: nutrition and agricultural intensification.

The first factor is nutrition, specifically the availability of forage (nectar and pollen sources) within the landscape. Foraging by pollinators is demonstrably affected by the land use composition around them [60,61]. Honeybees are generalists, employing a patch-based foraging strategy [62]. Landscape heterogeneity is directly related to the amount, richness and diversity of pollen that bees forage [63], and honeybees benefit from more diverse, heterogeneous landscapes [25,64].

All animals, even invertebrates, have an optimal diet that maximizes their fitness (or health), known as the “intake target” [65]. These diets are typically referred to in terms of a protein: carbohydrate ratio; research indicates that diets with an excess of protein (P:C ratio >5:1) can result in significantly shortened lifespans [66]. More moderate P:C ratios (~2:1) are generally associated with enhanced resistance to bacterial and viral pathogens. When given a choice of what to eat, insects are able to adapt what they eat depending on their state of health [67]. Honeybees are specially adapted to forage over a huge range (up to 10 km on a single flight), selecting what they bring back to the hive based on the colony’s fitness. However, if this range is saturated with a single floral species, as can be the case in some agricultural land, then this choice may be taken away from the foragers, hindering their ability to reach their preferred intake target.

The link between nutritional diversity and richness and pollinator fitness is well established; high quality diets have benefits to immune responses, reproduction and adult survival [68]. Although there is a lack of direct experimental data supporting dietary adaptation in honeybees, our current understanding is that honeybees that are under duress from pests or pathogens will consume a higher protein diet, whereas healthy bees may eat higher carbohydrate diets to enable greater exploitation of their environment. The nutritional status of honeybee populations has been shown to vary consistently with landscape type, with, for example, urban and forest environments being positive for honeybee nutrition [25,64,69]. In the UK, rural landscapes are dominated by agricultural land uses. Here, honeybees are most important for human well-being given their role in providing pollination services, yet these are sites of significant pressure on pollinators [70]. Agricultural habitats provide significant but time-limited nutritional resources, with extensive areas of single-crop arable farming flowering at once but only for a short time, negatively impacting nutritional diversity [71]. 

The second landscape-scale factor is also a result of agricultural intensification: the effects on honeybee health of pesticide use. The breadth and scope of knowledge on the damaging effects of chemical pesticides on honeybees is remarkable. Not only do insecticidal pesticides cause significant damage to honeybees [72], we are now beginning to appreciate the effects of herbicides and fungicides [73,74]. Given the volumes of fungicides and herbicides applied in agricultural landscapes, it is likely we are only beginning to appreciate the importance of the results of this emergent research. While beekeepers are key stakeholders responsible for ensuring bee health [75], and have historically provided evidence of pesticide risks to bees [26], in practice agrochemical risk and application is assessed and controlled by scientists and farmers, who may not be assessing bee health in the same way as beekeepers, or have the same priorities in land management [44,54]. 

## 6. Towards an Integrated Approach to Honeybee Health

We have shown how honeybees live in a complex social-ecological system, influenced by highly varied forms of human management, from the hive- to landscape-scale. The One-Health concept can effectively bridge these different sectors and scales, focusing as it does on interdisciplinary approaches to understanding health. In this paper, we have considered research from animal and human geography, microbiology, ecology, entomology, nutrition and many other disciplines (Figure A1). When considered together, the result is a clearer understanding of the many factors that influence honeybee health. Zinsstag [76] argue for a systemic approach to understanding health within social–ecological systems at a time of global change. There are many other factors that influence bee health, which we have not explored here, especially seasonal and climatic changes. Disruption of seasonal patterns can have serious knock-on effects to pollinators, particularly through changing forage plant phenology [59]. As we have previously discussed, disruption to foraging on a landscape scale can have significant negative consequences for honeybee health; complicating this, disruption on a temporal scale through climate change would clearly have even further negative consequences. The factors we have discussed will continue to shift and interact with other influences, forcing honeybees to adapt and change to maintain good health. To maintain honeybee health, these adaptations must be understood and supported by those most closely involved with their daily management, i.e. beekeepers and bee inspectors.

Beekeepers’ knowledge is applied at the interface between landscape and hive-scale factors, and has a great potential role in supporting a One-Health approach. One way to understand this role is as a form of citizen science (CS). CS projects have become an important method of increasing engagement between practitioners and researchers, and enriching policy [77,78]. The dominant model of CS with beekeepers was characterised by experiments that were designed by formal scientists, with beekeepers incorporated as data collectors, and requested to submit fairly basic information (e.g., annual colony losses, honey production) [43,79]. Several of these projects have been run for many years, generating key baseline information (BIP, BeeBase, and CoLOSS). As these projects have evolved, further questions have been incorporated into the surveys, exploring the knowledge and practices of beekeepers, and how these influence colony survival [49]. 

Alternative CS models emphasise collaborative and co-created approaches, which allow for a higher degree of scientific engagement with beekeepers’ knowledge [80]. Volunteers and other ‘amateurs’ have a significant role to play in environmental monitoring; unfortunately, this is frequently underutilised due to their being outside formal scientific communities [81]. Beekeeping is often an intergenerational practice, carried out by highly skilled, environmentally observant individuals. This, coupled with the habit of keeping hive records, and notes of other relevant factors to bee health and productivity, can generate detailed information on landscape-level changes to forage. Sufficiently detailed records also represent a bank of data on in-hive factors, including medication histories, queen breeding histories, and more. These observations can contribute to a multiple evidence based approach to policy development [55].

The UK’s exit from the European Union will bring changes to environmental and agricultural policy, with potentially far-reaching effects on honeybee health. Reflecting on how best to develop new policies that support honeybee health is important, especially given the investment to date in honeybee health in the UK through the Insect Pollinator Initiative and more [82]. While landscape-scale factors are beyond most veterinary medical research, honeybees, much like all wild animals, operate at this scale. Historically, the EU has used agri-environment schemes that support the development of rural areas, to reverse the decline in pollinator biodiversity and their associated agro-ecosystems, and to protect biodiversity and ecosystem function. The Common Agricultural Policy (CAP) supported localised greening measures, such as crop diversification, protection of permanent grassland from conversion to arable and the implementation of ecological focus areas. These aim to improve pollinator resilience and help combat losses [3]. Reviews of the overall success of CAP have suggested the results have been limited due to the nature of payments focusing on land ownership, rather than environmental intent. Consequently, the effects on landscapes have been less than effective [83]. New land management policies should incorporate factors supporting honeybee health at all scales.

The UK Government Agriculture Bill 2020 is centered on the new Environmental Land Management system (ELMs). This will determine subsidies based on “public pay for public goods” (UK Gov., 2020), adopting a natural capital valuation approach. When considering honeybees, due to the effective area to forage in and the context of the ‘pollinator movement hypothesis’ [84], cooperative habitat management at the landscape scale has more evident and immediate benefits to these mobile ecosystem service providers [85]. Future land management strategies should consider implementation at the inter-farm level, highlighting the importance of complementarity of resources. Practical impacts from national policy initiatives have consistently struggled to reach beyond localised effects [86]. Action to counter pollinator decline has frequently manifest in ways which are seen as piecemeal, and motivated by an effort to appeal to public interest, rather than fundamentally addressing landscape-scale challenges to bee health [87].

Cost-effectiveness in ecosystem conservation can be achieved through the implementation of multi-functionality or “stacking” services to maximise output from minimal input [88]. Applying a multi-functional ecosystem service framework may result in exponentially greater, synergistic and efficient use of limited resources [85,89], but equally, requires the understanding and approval of practitioners, and an incentive to elicit environmental change. Though lacking under CAP, ELMs may suitably address this necessity, though this proposal remains in the consultation phase and is unlikely to show demonstrable results for the near future.

There is a worrying dissonance between agricultural policies, and the ecological status of honeybees [90]. Wider research on conservation policy notes the importance, and benefits, of integrating the knowledge and concerns of disparate communities [91]. A top-down policy approach will not prove an effective pathway towards integrating a One-Health concept into bee health, and efforts at inclusion must be carefully assessed to assure they move beyond rhetoric [92]. The One-Health paradigm supports community-led, bottom-up approaches that link vets, beekeepers, landowners and policy makers. Such an approach has the capacity to engage with diverse factors affecting decision-making in animal husbandry, such as a sense of community responsibility, and a belief, or lack thereof, in individual capacity to affect change [93]. While this model has been gaining traction throughout the veterinary community, and raises important points about the transdisciplinary nature of contemporary zoonotic infections, there is a risk that this perspective still tends to overlook relational realities, as well as their socio-economic and cultural settings [20]. These are the locally specific, subjectively experienced factors that can give rise to differing interpretations of bee health and how best to ensure a colony’s wellbeing. 

The public and policy responses to honeybee decline in recent years reflect an awareness of the interwoven fates of humans and pollinators (albeit with some rhetorical misunderstandings, as noted by [94]). This awareness is emblematic of other interspecific global challenges that are being tackled via the One-Health approach [95]. A One-Health approach to bee health highlights the need to incorporate landscape-scale factors alongside other factors functioning at the hive-, and the human- scale. Unravelling their effects on honeybees requires research across many disciplines, including, but not limited to, nutrition, pollination ecology, microbiology, neuroscience, and agronomy. This emphasises that conceptualising “health”, simply as presence/absence of disease, is insufficient for understanding pollinator fitness (Table A1). Successfully applying a One-Health model to bee health will require active engagement with diverse and, at times, antagonistic parties, many of whose actions are driven by wider economic forces. If applied within the One-Health model, veterinary knowledge can make a strong contribution to reversing honeybee decline.

Honeybee health cannot be separated from the health of the environment that surrounds them [25]. A single health issue for honeybees, e.g., the presence/absence/treatment of a mite, can make an entire agricultural and food system dependent on their pollinating and honey-producing activities vulnerable [59]. Furthermore, this risks adopting an “*Apis*-centric” perspective on pollinator conservation. Though honeybees are highly efficient pollinators, they are by no means the only active player in this system [96]. Honeybee pathogens have also demonstrated the potential for zoonotic transfer to non-*Apis* pollinators when visiting the same flower patches [51]. Consequently, as well as considering honeybee health in a holistic manner, we must also consider knock-on impacts of treatment regimens and fitness assessments on to non-target species. Adopting a One-Health approach, deliberately seeking to bridge across the diverse sectors that affect bee lives, is therefore crucial.

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
