# Peer review of "A One-Health Model for Reversing Honeybee (Apis mellifera L.) Decline"

_vetsci, 2020, doi:10.3390/vetsci7030119_

Round 1

Reviewer 1 Report

Thank you for the opportunity to review this manuscript. Honeybee health is exceedingly important and this paper nicely reviews the idea that it requires a multidisciplinary approach.

My major concern is that this is in no way an "Article" which defined by this journal as "Original research manuscripts. The journal considers all original research manuscripts provided that the work reports scientifically sound experiments and provides a substantial amount of new information." The methods section mentions that the authors worked with and interviewed people. However, this study could not be repeated, as the questions are not given. In addition, no data are presented. There is no real research design, statistical analysis, or means for identifying scientific soundness. Thus, I am rejecting the paper in its present form due to this alone.  If the editor disagrees and thinks it is an appropriate "article" I would love to see it in print.

That said, if there were a different category in which to put this article, such as "editorial," "perspective," or "opinion piece" it would be very welcome. A One Health Approach is absolutely needed for reversing honeybee decline. I value the discussion of the various aspects that are being explored in an effort to help honeybees to thrive.

Specific, line by line comments:

40: The word "approach" belongs as the last word.

50-52: Awkward as written

85: health should be "healthy"

130-132: Needs a reference

160: Hives should be "colonies". These two words are not interchangeable and a "hive" cannot be healthy..

230: At the end of the sentence add "; nutrition and agricultural intensification" 

297-300: This is a long, run-on sentence

305: Is "Pollinator Initiatives" a formal group? If not, why are these words capitalized?

324: Check grammar

339: Delete "a" in front of community-led

350: I am not sure of the appropriateness of the word "trans-species"

356: I feel that this sentence is an opportunity to discuss the need to define what HEALTHY means for honeybees. For example, you could delete "why bees experience poor health" and replace it with something like "whether or not a colony is healthy."

Author Response

Comments and Suggestions for Authors

Thank you for the opportunity to review this manuscript. Honeybee health is exceedingly important and this paper nicely reviews the idea that it requires a multidisciplinary approach.

My major concern is that this is in no way an "Article" which defined by this journal as "Original research manuscripts. The journal considers all original research manuscripts provided that the work reports scientifically sound experiments and provides a substantial amount of new information." The methods section mentions that the authors worked with and interviewed people. However, this study could not be repeated, as the questions are not given. In addition, no data are presented. There is no real research design, statistical analysis, or means for identifying scientific soundness. Thus, I am rejecting the paper in its present form due to this alone.  If the editor disagrees and thinks it is an appropriate "article" I would love to see it in print.

That said, if there were a different category in which to put this article, such as "editorial," "perspective," or "opinion piece" it would be very welcome. A One Health Approach is absolutely needed for reversing honeybee decline. I value the discussion of the various aspects that are being explored in an effort to help honeybees to thrive.

*** We agree with the reviewer on this point. At submission we had agreed with the editor of this special issue of Veterinary Sciences (Giovanni Cilia) that the manuscript would be a “Perspectives” or “Opinion Piece”. The submission portal did not allow for this option, however, we are in discussions with the editor currently to resolve this.

Specific, line by line comments:

40: The word "approach" belongs as the last word.

*** Agreed and amended.

50-52: Awkward as written

***Amended to: “Although the focus has been on zoonotic diseases, the One Health approach emphasises a holistic understanding to tackle challenges [13], and is therefore highly relevant for discussions about honeybee health.”

85: health should be "healthy"

*** Agreed and amended.

130-132: Needs a reference

*** Agreed and amended: new reference
36. Perry, C.J.; Søvik, E.; Myerscough, M.R.; Barron, A.B. Rapid behavioral maturation accelerates failure of stressed honey bee colonies. Proc. Natl. Acad. Sci. U. S. A. 2015, doi:10.1073/pnas.1422089112.

160: Hives should be "colonies". These two words are not interchangeable and a "hive" cannot be healthy..

*** Agreed and amended.

230: At the end of the sentence add "; nutrition and agricultural intensification" 

*** Agreed and amended. 

297-300: This is a long, run-on sentence

*** Agreed, sentence now reads:
“This, coupled with the habit of keeping hive records, and notes of other relevant factors to bee health and productivity, can generate detailed information on landscape-level changes to forage. Sufficiently detailed records also represent a bank of data on in-hive factors, including medication histories, queen breeding histories, and more.” 

305: Is "Pollinator Initiatives" a formal group? If not, why are these words capitalized?

*** Amended to: "Insect Pollinator Initiative”

324: Check grammar

***Amended to: “Practical impact of national policy initiatives has been consistently limited to localised effects [85]

339: Delete "a" in front of community-led

***Agreed and amended

350: I am not sure of the appropriateness of the word "trans-species"

***Amended to: “interspecific”

356: I feel that this sentence is an opportunity to discuss the need to define what HEALTHY means for honeybees. For example, you could delete "why bees experience poor health" and replace it with something like "whether or not a colony is healthy."

***Amended to: “This emphasises that conceptualising “health”, simply as presence/absence of disease is insufficient to understanding pollinator fitness.”

Reviewer 2 Report

The manuscript is very sound and well written although I have the impression it is rather a review. The information is nicely integrated in the single sections but in my opinion there are some points missing.

Like the authors recognized and mentioned in their manuscript, in fact, climate change is one of the most important factors with a huge impact on honeybees (and their interactions) on all relevant scales (e.g. agriculture, colony survival, pathogen/ parasite spread, interacting species …). This should be discussed.

The interpretations and conclusion of the manuscript are a way to restricted and should be expanded to a brother view on the impact off honeybees in landscapes and the ecosystem. Honeybees do not just have a positive impact. Besides negative fitness effects and displacement of other pollinators (the authors already mentioned it with one citation in section 6) honeybees also may be vectors of pathogens and driving the spread of diseases specifically in an intensified honeybee economy. I have the feeling that the authors already pointing into this direction in section 4 but do not get specific about it.

Box 1: The heading of the box should be changed as the authors also mention the tracheal mite Acarapis woodi. Further, there is not much evidence that F. perrara is a pathogen, rather a mutualist or commensal (the authors mention it a couple of lines before). Is it a mistake?

Author Response

The manuscript is very sound and well written although I have the impression it is rather a review. The information is nicely integrated in the single sections but in my opinion there are some points missing.

Like the authors recognized and mentioned in their manuscript, in fact, climate change is one of the most important factors with a huge impact on honeybees (and their interactions) on all relevant scales (e.g. agriculture, colony survival, pathogen/ parasite spread, interacting species …). This should be discussed.

*** Agreed, although we made the clear indication that we did not intend to discuss climate change in the original submission of the manuscript, we have amended this in the new submission.

L282-287: There are many other factors that influence bee health which we have not explored here, especially seasonal and climatic changes. Disruption of seasonal patterns can have serious knock-on effects to pollinators, particularly through changing forage plant phenology [59]. As we have previously discussed, disruption to foraging on a landscape scale can have significant negative consequences for honeybee health; complicating this, disruption on a temporal scale through climate change would clearly have even further negative consequences. 

The interpretations and conclusion of the manuscript are a way to restricted and should be expanded to a brother view on the impact off honeybees in landscapes and the ecosystem. Honeybees do not just have a positive impact. Besides negative fitness effects and displacement of other pollinators (the authors already mentioned it with one citation in section 6) honeybees also may be vectors of pathogens and driving the spread of diseases specifically in an intensified honeybee economy. I have the feeling that the authors already pointing into this direction in section 4 but do not get specific about it.

***Agreed, we have revised the discussion of negative effects of honeybees on non-Apis pollinators in section 6, and have included a “sign post” directing readers to this in section 4. Revision:

L197-198: “...or even to non-Apis pollinators [51] (see Towards an Integrated Approach to Honeybee Health).”

L:374-379:  Furthermore, this risks adopting an “Apis-centric” perspective on pollinator conservation. Though honeybees are highly efficient pollinators, they are by no means the only active player in this system [96]. Honeybees pathogens have also demonstrated the potential for zoonotic transfer to these non-Apis pollinators when visiting the same flower patches [51]. Consequently, as well as considering honeybee health in a holistic manner, we must also be considering knock-on impacts of treatment regimens and fitness assessments on to these non-target species. 

Box 1: The heading of the box should be changed as the authors also mention the tracheal mite Acarapis woodi. Further, there is not much evidence that F. perrara is a pathogen, rather a mutualist or commensal (the authors mention it a couple of lines before). Is it a mistake?

***Box 1 has been amended to Table 1 in the revised version of the manuscript, and addresses this concern.

Round 2

Reviewer 1 Report

I am very much looking forward to seeing this in print, assuming the issue of the type of paper is resolved. 

"At submission we had agreed with the editor of this special issue of Veterinary Sciences (Giovanni Cilia) that the manuscript would be a “Perspectives” or “Opinion Piece”. The submission portal did not allow for this option, however, we are in discussions with the editor currently to resolve this."

Thank you for this work and I look forward to its publication.